# Detecting Intrathoracic Airway Closure during Prehospital Cardiopulmonary Resuscitation Using Quasi-Static Pressure–Volume Curves: A Pilot Study

**DOI:** 10.3390/jcm13144274

**Published:** 2024-07-22

**Authors:** Maxim Vanwulpen, Arthur Bouillon, Ruben Cornelis, Bert Dessers, Saïd Hachimi-Idrissi

**Affiliations:** 1Emergency Department, Ghent University Hospital, Corneel Heymanslaan 10, 9000 Ghent, Belgiumsaid.hachimiidrissi@ugent.be (S.H.-I.); 2Faculty of Medicine and Health Sciences, Ghent University, Sint-Pietersnieuwstraat 25, 9000 Ghent, Belgium; 3Faculty of Medicine and Pharmacy, Free University Brussels, 1090 Brussels, Belgium

**Keywords:** cardiac arrest, cardiopulmonary resuscitation, ventilation

## Abstract

**Background:** Intrathoracic airway closure frequently occurs during cardiac arrest, possibly impairing ventilation. Previously, capnogram analysis was used to detect this pathophysiological process. In other populations, quasi-static pressure–volume curves obtained during constant low-flow inflations are routinely used to detect intrathoracic airway closure. This study reports the first use of quasi-static pressure–volume curves to detect intrathoracic airway closure during prehospital cardiopulmonary resuscitation. **Methods:** Connecting a pressure and flow sensor to the endotracheal tube enabled the performance of low-flow inflations during cardiopulmonary resuscitation using a manual resuscitator. Users connected the device following intubation and performed a low-flow inflation during the next rhythm analysis when chest compressions were interrupted. Determining the lower inflection point on the resulting pressure–volume curves allowed for the detection and quantification of intrathoracic airway closure. **Results:** The research device was used during the prehospital treatment of ten cardiac arrest patients. A lower inflection point indicating intrathoracic airway closure was detected in all patients. During cardiac arrest, the median pressure at which the lower inflection point occurred was 5.56 cmH20 (IQR 4.80, 8.23 cmH20). This value varied considerably between cases and was lower in patients who achieved return of spontaneous circulation. **Conclusions:** In this pilot study, quasi-static pressure–volume curves were obtained during prehospital cardiopulmonary resuscitation. Intrathoracic airway closure was detected in all patients. Further research is needed to determine whether the use of ventilation strategies to counter intrathoracic airway closure could lead to improved outcomes and if the degree of airway closure could serve as a prognostic factor.

## 1. Introduction

In recent years, intrathoracic airway closure has been suggested to occur following cardiac arrest [1]. Chest compressions cause lung volume reduction below the end-expiratory volume, resulting in the closure of the intrathoracic airways, possibly leading to impaired gas exchange [2,3,4]. In a human cadaver model, varying amounts of positive end-expiratory pressure (PEEP) were shown to counteract this pathophysiological process [5]. The hemodynamic effects of PEEP during CPR remain debated. Animal studies have suggested that PEEP could impair cardiac output, it is unclear if this is also a concern in humans [6,7].

Analyzing the amplitude of oscillations in the capnogram generated by chest compressions has been proposed to detect intrathoracic airway closure during CPR [2,6]. Using this method, significant airway closure was reported to occur in 35% of out-of-hospital cardiac arrest (OHCA) cases [8]. The ability of this method to detect small or varying levels of intrathoracic airway closure is unknown [5].

Intrathoracic airway closure has been observed during acute respiratory distress syndrome and in healthy, mechanically ventilated patients undergoing general anesthesia [4,9]. Determining the presence of a lower inflection point (LIP) on total respiratory system pressure–volume curves obtained during constant low-flow inflations is a commonly used quasi-static method to detect intrathoracic airway closure in mechanically ventilated patients [9,10,11,12,13]. A low-flow inflation is used to minimize resistive effects on measured airway pressure. The influence of the resistance factor on low-flow pressure–volume curves is not clinically relevant if the administered flow is less than 10 L/min [11,14].

The LIP is defined as the pressure corresponding to the intersection of two lines representing minimal and maximal compliance on the pressure–volume curve. This value is influenced by the amount of potentially recruitable lung and regional mechanical differences in pulmonary tissue (e.g., heterogeneous pulmonary injury) and chest wall characteristics [11]. A higher LIP is associated with more extensive intrathoracic airway closure [12]. In healthy patients, an LIP is not observed during normal ventilation [13]. Setting PEEP above the observed LIP has been shown to minimize or avoid intrathoracic airway closure in other populations [9].

In this pilot study, we report the first use of low-flow pressure–volume curves to detect and quantify intrathoracic airway closure during prehospital cardiopulmonary resuscitation.

## 2. Materials and Methods

### 2.1. Materials

The research device consisted of a pressure sensor (CPT2500, WIKA [Klingenberg, Germany] measurement range −102–102 cmH20, accuracy 0.1%, update time 1 ms), an airflow sensor (SFM3200, Sensirion [Stäfa, Swiss] measurement range −100–250 L/min, accuracy 2%, update time 1 ms), and a tablet computer (Surface Go 2, Microsoft [Redmond, WA, USA]).

To provide airway pressure measurements, the pressure sensor was connected to a pressure-transducing catheter, entering the endotracheal tube through a custom coupling piece and fitting on the endotracheal tube. A respiratory filter separated the airflow sensor from the coupling piece. A manual resuscitator (BVM resuscitator, Intersurgical [Wokingham, GB]) was connected to the other side of the airflow sensor (Figure 1).

If airflow was detected, data recording started automatically. Measured flow and information on the use of the device were shown on the screen (Figure 1). Sensirion USB sensor viewer V2.9 and Wika USBsoft 2500 V1.4 were used. A monitor defibrillator (X Series, Zoll [Chelmsford, MA, USA]) with real-time chest compression feedback was used. A mechanical chest compression system (LUCAS 2, Stryker [Kalamazoo, MI, USA]) was available. Patients were intubated using a cuffed endotracheal tube, and the size was determined at the physician’s discretion.

### 2.2. Methods

This was a single-center, prospective observational study. The research device was planned to be used in a convenience sample of ten adult OHCA cases treated by the prehospital medical team of the Ghent University Hospital Emergency Department (responding on average to 100 OHCA cases per year). Patients were included at the discretion of the treating physician. If flow exceeded 10 L/min during the constant low flow maneuver, patients were excluded from the analysis. No a priori power analysis was performed in this pilot study. Ethical approval was obtained from the Ghent University Hospital institutional review board, informed consent was obtained from the patient or the patient’s next of kin (reference number EC/2008/025/AM02). Funding for this study was obtained from the Zoll Foundation (internal grant number KW/2191/NKU/001/013).

During CPR, the research device was connected to the ventilatory circuit of the patient immediately following endotracheal intubation. In accordance with current resuscitation guidelines, chest compressions were paused every two minutes to analyze the cardiac rhythm and determine whether a return of spontaneous circulation had occurred. During this first short interruption in chest compressions following intubation, users were asked to perform a manual low flow inflation guided by the flow sensor (aiming to provide a constant flow < 10 L/min) until chest compressions were resumed or until a pulse was detected.

### 2.3. Analysis

Following the use of the research device, data was imported and synchronized in an Excel macro. The inspired volume during each inflation was calculated, and a pressure–volume curve was generated. Lines indicating minimal and maximal compliance were added. The pressure at which these lines intersected was calculated, indicating the LIP (Figure 2).

The LIP was reported using descriptive statistics. The distribution of the collected data was assessed visually. Pearson’s correlation analysis was used to determine the relationship between the time interval during which CPR was performed prior to performing low-flow inflation and the calculated LIP. Student’s *t*-test was used to compare the mean LIP when ROSC did or did not occur. The significance level was set at 0.05.

## 3. Results

The research device was available for use between October 2022 and April 2023. No patients were excluded from the analysis. Measurements were performed during the prehospital treatment of ten OHCA patients. Eight patients were male and the median age was 72 years (IQR 55, 80). The median time between cardiac arrest and manual low flow inflation was 25 min (IQR 20, 30). Summarized patient and treatment characteristics are shown in Table 1.

A LIP was detected in all patients. The median LIP was 5.56 cmH20 (IQR 4.80, 8.23 cmH20). The calculated LIP and the time interval during which CPR was performed prior to low-flow inflation followed a normal distribution.

Univariate analysis showed a statistically significant, strong correlation between the time interval during which CPR was performed prior to low-flow inflation and the calculated LIP (r_s_(8) = 0.874, *p* < 0.01). This relationship is visualized in Figure 3.

When ROSC was achieved, the calculated LIP was significantly lower (mean 3.15 cmH20, standard deviation 1.92 cmH20) compared to when ROSC was not achieved (mean 6.53 cmH20, standard deviation 2.10 cmH20; t(8) = 2.39, *p* = 0.04).

## 4. Discussion

In this pilot study, quasi-static low-flow pressure–volume curves were used to detect intrathoracic airway closure during prehospital CPR. The novel method introduced in this study enhances our ability to study airway closure during cardiac arrest.

A LIP was observed in all patients. Calculated LIPs varied considerably between cases, indicating different degrees of intrathoracic airway closure. This was also observed in previous studies using capnogram analysis [6]. If a longer time had elapsed between the start of CPR and low-flow inflation, the LIP was higher. Intrathoracic airway closure possibly progressively increases during CPR. Performing repeated measurements in the same patient could enable further exploration of this hypothesis.

In cases during which ROSC was achieved, the calculated LIP was lower. Possibly, the occurrence of intrathoracic airway leads to reduced oxygenation and ventilation, negatively affecting the chance of obtaining ROSC. Alternatively, more extensive intrathoracic airway closure may occur in patients who have other factors associated with a lower likelihood of achieving ROSC (e.g., pre-existing cardiopulmonary disease, longer duration of CPR). The number of patients included in this pilot study was too small to adequately assess these relationships. Further studies evaluating the value of the LIP as a possible prognostic factor should be performed.

In a previous human cadaver study, the use of PEEP was shown to counteract the occurrence of intrathoracic airway closure [5]. Further development of the device introduced in this study could allow real-time feedback on the occurrence of intrathoracic airway closure to be provided. A randomized controlled trial on the use of individualized levels of PEEP, determined based on real-time calculated LIPs during CPR, could be conducted.

Setting PEEP above the observed LIP has been reported to minimize or avoid intrathoracic airway closure in other patient populations [9]. Caution should be applied when employing this strategy. Whilst the occurrence of a LIP is associated with intrathoracic airway closure, recruitment and airway closure have been observed to occur at pressures both below and above the LIP in acute respiratory distress syndrome. This is likely caused by heterogeneous pulmonary injury, which could also exist during cardiac arrest [12]. Additionally, two studies in acute respiratory distress syndrome patients failed to show clinical benefits when using higher or titrated levels of PEEP [15,16].

The hemodynamic effects of applying PEEP during CPR in humans are unknown. Animal studies have suggested that the use of PEEP could cause a decrease in cardiac output. No human data are currently available [6,7].

### Limitations

The stress of performing additional tasks whilst managing cardiac arrest patients likely limited the use of the research device, leading to a relatively long study period to recruit a limited number of patients. Care was taken to simplify the use of the device and minimize the risk of selection bias.

Intrathoracic airway closure has been suggested to be caused by chest compressions, airway edema, and loss of chest wall muscle tone [2,3]. Pre-existing pneumopathy could have also influenced the results. The number of patients included in this pilot study was too small to assess which factors influenced the occurrence of intrathoracic airway closure or determine whether intrathoracic airway closure occurs in all cardiac arrest patients. The external validity of this pilot study is limited. Further studies, including more patients and performing repeated measurements in the same patient, should be performed.

The addition of a custom coupling piece, pressure transducing catheter, endotracheal tube, and airflow sensor to the ventilatory circuit generated additional airway resistance. This resistance was constant during the entire low flow inflation and was unlikely to influence the occurrence of the LIP.

To avoid interruptions in CPR, low-flow inflations were performed manually. Small variations in flow and dynamic airway resistance likely occurred during these inflations, possibly impacting the LIP. Previous research has shown the influence of the resistance factor on pressure–volume curves to be clinically irrelevant if the flow is lower than 10 L/min [11,14]. However, formal validation and calibration of the proposed method and device should be pursued in future studies.

Capnograms were not collected during this study, and no comparison could be made between the capnography-based method of detecting intrathoracic airway closure and the novel method. No blood gas analyses were performed in this study, and the effectiveness of oxygenation and ventilation during CPR was not known. Using simultaneous pulse-oximetry in future studies could provide additional information on oxygenation during cardiac arrest.

## 5. Conclusions

In this pilot study, quasi-static pressure–volume curves were obtained during prehospital CPR. Intrathoracic airway closure was detected in all patients and varied considerably between cases. Further research is needed to determine whether ventilation strategies to counter intrathoracic airway closure (e.g., individualized levels of PEEP) could lead to improved outcomes and if the degree of airway closure could serve as a prognostic factor during CPR.

## Figures and Tables

**Figure 1 jcm-13-04274-f001:**
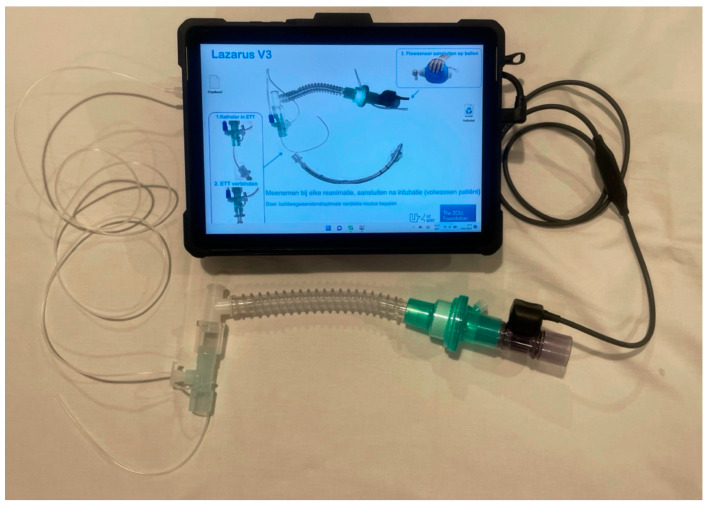
Research device—pressure-transducing catheter and custom coupling piece on the left, flow sensor on the right.

**Figure 2 jcm-13-04274-f002:**
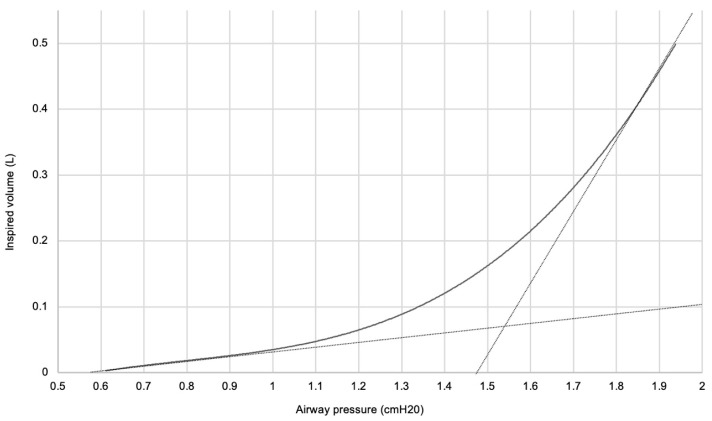
Pressure–volume curve of a manual low-flow inflation—dotted lines indicate minimal and maximal compliance, and the pressure at which these lines intersect is defined as the lower inflection point.

**Figure 3 jcm-13-04274-f003:**
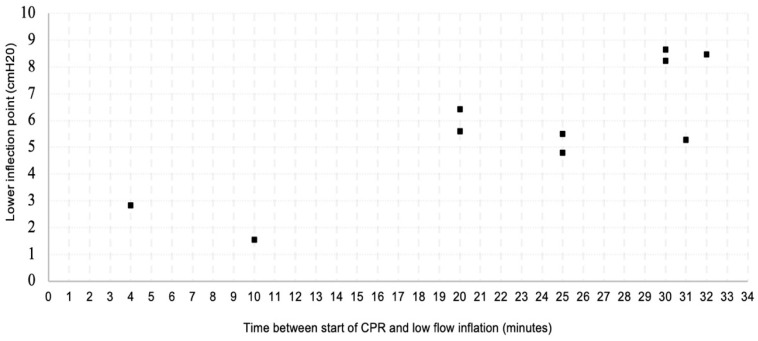
Scatter plot—time interval between the start of CPR and low-flow inflation is shown on the horizontal axis, calculated lower inflection point is shown on the vertical axis.

**Table 1 jcm-13-04274-t001:** Summarized patient and treatment characteristics.

Age	72 years (IQR 55, 80)
Bystander cardiopulmonary resuscitation	6/10
Return of spontaneous circulation	3/10
Cardiac catheterization	3/10
Survival to hospital discharge	2/10
Type of chest compressions performed	
Manual chest compressions	8
Mechanical chest compressions	2
First monitored rhythm	
Asystole	6
Ventricular fibrillation	2
Ventricular tachycardia	1
Pulseless electrical activity	1
Presumed cause of cardiac arrest	
Cardiac ischemia	4
Pulmonary embolism	2
Aortic dissection	1
Sepsis	1
Intracranial bleed	1
Asphyxia	1
Sex	
Male	8
Female	2

## Data Availability

Data are available upon request from the authors.

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
