# Peer review of "Detecting Intrathoracic Airway Closure during Prehospital Cardiopulmonary Resuscitation Using Quasi-Static Pressure–Volume Curves: A Pilot Study"

_jcm, 2024, doi:10.3390/jcm13144274_

Round 1
Reviewer 1 Report
Comments and Suggestions for Authors
Dear authors,
In the brief report entitled “Detecting intrathoracic airway closure during prehospital cardiopulmonary resuscitation using quasi-static pressure-volume curves: a pilot study” you have presented an interesting technique to monitor airway closure, a possible life-threatening condition with implications on return of spontaneous circulation (ROSC) and survival in patients needing cardiopulmonary resuscitation (CPR).
I salute the novelty of the technique, which seems easy to adapt in a very stressful setting, that of CPR. The relevant information to understand the method per se is found in the background. The method is explained into detail and the first results on 10 out of hospital cardiac arrest patients (OHCA) are shown. Notably, with a rate of ROSC of 30%, the collective seems to have very bad outcomes.
The discussion needs some improvements. I ask the authors to address the following points:
Major comments:
1. What are the clinical implications of this measurement? Can it aid the CPR process, for instance by improving ventilation processes? Or is it meant to forecast ROSC or survival, respectively? If yes, would the intrathoracic airway closure as a binomial variable be the leading prognostic marker or might the lower inflection point (LIP) act as a marker? Or is it maybe only of research application without ever finding the way to bedside.
2. After discussing the clinical relevance of the device, the authors should provide more insight into what the future applications should be. Where should the device be used? What are future aspects to consider in researching this method. Is for instance a randomized controlled trial possible, to aid CPR with and without LIP measurement?
3. Please provide some more details to the OHCA patients (cause of OHCA, whether catheterizations were performed, survival, neurological outcomes). Did diagnosing airway closure have any effect on the decision to stop CPR in patients without ROSC?
Minor:
1. Does the size of the intubation tubus affect somehow the measurement?
2. Is there any statistically significant relationship between the variables shown in figure 3?
3. Did patients of next of kin consent to the study?
Reviewer 2 Report
Comments and Suggestions for Authors
The aim of this study by Vanwulpen et al. was to evaluate intrathoracic airway closure during pre-hospital cardiopulmonary resuscitation (CPR) using quasi-static pressure-volume curves generated using data obtained from the pressure and flow sensor connected to the endotracheal tube during CPR. Determining the presence of a lower inflection point (LIP) in the pressure-volume curves was the tool used to detect and quantify airway closure.
According to one of the studies mentioned in the introduction (Lesimple et al.), intrathoracic airway closure during CPR was observed in 35% of cardiopulmonary arrest cases, when capnography was used for analysis. However, in the study carried out, such closure was detected in 100% of the ten patients evaluated.
This difference may be due to the methodologies adopted by the studies, since the same phenomenon was assessed differently in these two studies, one using capnography and the other using the pressure/flow sensor. However, we cannot rule out the possibility of measurement bias, such as inadequate calibration of the instrument used. Alternatively, these patients may have had previous pneumopathy (e.g. asthma, COPD).
Also in the Introduction, the author of the study states that setting PEEP above the observed LIP minimizes or avoids intra-thoracic airway closure in other populations, citing the article by Hedenstierna et al. However, the clinical relevance of PEEP values is still very controversial in the literature. There are multicenter randomized clinical trials, such as the National Heart, Lung, and Blood Institute (NHLBI) ARDS Clinical Trials Network, published in 2004, as well as the Alveolar Recruitment for Acute Respiratory Distress Syndrome Trial (ART) Investigators, from 2017, which compare different PEEP values in the management of respiratory distress syndrome (ARDS), showing no benefit for different clinical outcomes. Although these clinical trials addressed a different population (ARDS vs. CRP patients) from the Vanwulpen study, the latter could have mentioned the dubious impact of determining PEEP in different clinical contexts.
The study is single-center and small, with a sample of ten patients. It is a pilot study, with the initial focus on the feasibility of the technique. Therefore, the external validity and inference of the result obtained over the entire population must be questioned as its limitations.
Although the results may be limited, I would suggest using a pulse oximeter (SpO2) for future studies.
In the Results, in addition to the baseline characteristics of the patients, the time interval between the start of CPR and the LIP is shown graphically (Figure 3). And for greater richness in the Discussion, another correlation could be added: the LIP value and the return to spontaneous circulation (ROSC). According to the study, only three of the ten patients returned to spontaneous circulation, and knowledge of the relationship between the LIP value and the ROSC would be interesting for evaluating the clinical impact of ventilation strategies to prevent intrathoracic airway closure (e.g. PEEP), as well as indicating new directions for future research, as concluded in this study by Vanwulpen et al.
1) Brower RG, Lanken PN, MacIntyre N, Matthay MA, Morris A, Ancukiewicz M, Schoenfeld D, Thompson BT; National Heart, Lung, and Blood Institute ARDS Clinical Trials Network. Higher versus lower positive end-expiratory pressures in patients with the acute respiratory distress syndrome. N Engl J Med. 2004 Jul 22;351(4):327-36. doi: 10.1056/NEJMoa032193. PMID: 15269312.
2) Writing Group for the Alveolar Recruitment for Acute Respiratory Distress Syndrome Trial (ART) Investigators; Cavalcanti AB, Suzumura ÉA, Laranjeira LN, Paisani DM, Damiani LP, Guimarães HP, Romano ER, Regenga MM, Taniguchi LNT, Teixeira C, Pinheiro de Oliveira R, Machado FR, Diaz-Quijano FA, Filho MSA, Maia IS, Caser EB, Filho WO, Borges MC, Martins PA, Matsui M, Ospina-Tascón GA, Giancursi TS, Giraldo-Ramirez ND, Vieira SRR, Assef MDGPL, Hasan MS, Szczeklik W, Rios F, Amato MBP, Berwanger O, Ribeiro de Carvalho CR. Effect of Lung Recruitment and Titrated Positive End-Expiratory Pressure (PEEP) vs Low PEEP on Mortality in Patients With Acute Respiratory Distress Syndrome: A Randomized Clinical Trial. JAMA. 2017 Oct 10;318(14):1335-1345. doi: 10.1001/jama.2017.14171. PMID: 28973363; PMCID: PMC5710484.
Comments on the Quality of English LanguageND
